# Evaluation of Celiac Disease by Minimally Invasive Biomarkers in a Spanish Pediatric Population

**DOI:** 10.3390/ijerph19095020

**Published:** 2022-04-20

**Authors:** Julia María Cabo del Riego, María Jesús Núñez Iglesias, Carmen García-Plata González, José Paz Carreira, Tamara Álvarez Fernández, Ana Dorado Díaz, Noa Villar Mallo, Manuel Penedo Pita, Silvia Novío Mallón, Lola Máiz Suárez, Manuel Freire-Garabal Núñez

**Affiliations:** 1Clinical Analysis Laboratory, Department of Inmunology, Hospital Universitario Lucus Augusti, 27003 Lugo, Spain; tamara.alvarez.fernandez@sergas.es (T.Á.F.); noa.villar.mallo@sergas.es (N.V.M.); manuel.penedo.pita@sergas.es (M.P.P.); lola.maiz.suarez@sergas.es (L.M.S.); 2Doctoral Programme in Medicine Clinical Research, International PhD School of the University of Santiago de Compostela (EDIUS), 15782 Santiago de Compostela, Spain; 3Department of Psiquiatry, Radiology, Public Health, Nursing and Medicine, University of Santiago de Compostela, 15705 A Coruña, Spain; silvia.novio@usc.es; 4SNL Laboratory, School of Medicine and Dentistry, 15782 A Coruña, Spain; manuel.freire-garabal@usc.es; 5Pediatric Gastrohepathology Unit of Hospital Universitario Lucus Augusti, 27003 Lugo, Spain; carmen.garcia-plata.gonzalez@sergas.es (C.G.-P.G.); jpazcarreira@hemogalicia.com (J.P.C.); 6Department of Statistical Health Counselling, Junta de Castilla y León, 47001 Valladolid, Spain; ana.dorado@jcyl.es; 7Department of Pharmacology, University of Santiago de Compostela, 15705 A Coruña, Spain

**Keywords:** non-invasive biomarkers, celiac disease, ESPGHAN, diagnosis, antibodies, pediatric age

## Abstract

Background: The diagnosis of celiac disease (CD) has been substantially improved with the availability of highly sensitive CD-specific IgA-TG2, Ig-GDP, and IgA-EMA. The European Society for Pediatric Gastroenterology, Hepatology, and Nutrition (ESPGHAN) published (2012) and updated (2020) diagnostic criteria for CD in order to simplify CD diagnosis and to avoid biopsies in selected patients. Methods: A prospective study including 5641 pediatric patients (0–16 years old) from January 2012 to January 2019 was performed. CD diagnosis was made according to the ESPGHAN algorithm. The objective of this study was to evaluate the utility of biomarkers and the relationship between TGA-IgA and EMA titers. Results: CD diagnoses were confirmed in 113 patients, 110 were IgA-TG2-positive and 3 (2.7%) had IgA deficiency. The diagnosis was made by serologic tests in 95 (84.1%) patients. Only 18 (15.9%) patients underwent intestinal biopsy. We obtained 100% concordance between IgA-EMA and positive results for IgA-TG2 ≥ 10 ULN with IgA-EMA antibody titer ≥ 1:80. Conclusions: This study provides evidence of a positive correlation between IgA-TG2 antibody serum levels and IgA-EMA. The diagnosis could be guaranteed with strict application of IgA-TG2 values ≥ 10 ULN (confirmed by subsequent testing) plus the serological response to the gluten-free diet (GFD).

## 1. Introduction

Celiac disease (CD) is considered one of the most common lifelong food-related disorders. This is an immune-mediated systemic disease triggered by gluten exposure with multifaceted clinical presentations such as gastrointestinal and/or extra-intestinal manifestations, CD-specific antibodies, and enteropathy, whose only effective treatment is a lifelong gluten-free diet (GFD) [1,2].

The diagnosis of the disease has been substantially improved with the availability of highly sensitive CD-specific tissue transglutaminase type 2 antibodies (IgA-TG2), IgG antibodies against deamidated gliadin peptides (IgG-DGP), and IgA anti-endomysial antibodies (IgA-EMA) [3,4], with correlations between severe atrophy of the duodenal villus and elevated IgA-TG2 and IgA-EMA titers [5,6,7,8,9].

One of the most important events of the last few years was the publication of the European Society of Pediatric Gastroenterology, Hepatology, and Nutrition (ESPGHAN) Guidelines for Diagnosing Coeliac Disease in 2012 [10]. These guidelines were focused on simplifying CD diagnosis and avoiding biopsy in selected patients. It was recommended that children and adolescents with CD-suggestive symptoms and IgA-TG2 ≥10 times the upper limit of normal (ULN) confirmed by IgA-EMA positivity, in a second serological test, as well as positivity for human leukocyte antigen (HLA) DQ2 or DQ8 haplotype, should be diagnosed without small bowel biopsy (SBB). In any case, the diagnosis had to be confirmed by serologic normalization after a GFD. Recently, the updated and expanded evidence-based guidelines were published [11].

After the publication of the 2012 ESPGHAN Guidelines, our research group started a study aiming to apply its guidelines in a wide pediatric population with suspected CD, from northwest Spain. Our objectives were: (a) to analyze the role of biochemical and genetic serological markers in order to reduce the number of biopsies performed, and (b) to establish serological outcomes after 2 years of a GFD.

## 2. Materials and Methods

### 2.1. Study Design and Subjects

We performed a prospective study on all newly diagnosed cases of childhood CD, according to the 2012 ESPGHAN algorithm, from January 2012 to January 2019.

In total, 5641 pediatric patients (0–16 years old) with manifestations of suspected CD or individuals who were still asymptomatic but at elevated risk of CD were studied. All the patients were referred to the Clinical Laboratory Service (University Hospital Lucus Augusti) by a pediatric gastroenterologist or by Galician Healthcare Service (SERGAS) community health centers in order to confirm the CD diagnosis. Pediatric patients were followed for up to 2 years.

### 2.2. Serological CD Diagnosis

All serological tests performed are accredited by the UNE-EN-ISO 15,189 standards for clinical laboratories (accreditation May 2011; reaccreditation 2013, 2015, 2017, 2019, and 2020).

At first, for symptomatic children [10], we determined the total IgA levels and IgA-TG2 antibodies. The decision to use IgA-TG2 as the first step was based on the high sensitivity and specificity of the test, its broad availability, and its low cost [10,11].

Since the possibility of false negatives for IgA-TG2 in children <4 years old is known (10), as additional evidence, we determined the IgG-GDP. In these cases, our choice was based on several reasons, namely that in some cases, it is the first marker to be positivized [12], as well as its availability in our laboratory and its low cost.

All serum samples were analyzed for total serum IgA using BNII nephelometry according to the manufacturer’s recommendations (Siemens BNII, Oststeinbek, Germany).

In the case of patients with selective IgA deficiency, we determined the IgG-GDP as the first step, and performed at least one additional IgG class test (anti IgG-TG2 and IgG-EMA). If these tests were positive, these patients were scheduled for biopsy.

IgA deficiency was determined according to age as follows: in patients aged <4 years old, it was calculated from a linear curve of pediatric IgA values [13], and for those aged ≥4 years, it was calculated according reference values (70 mg/dL) and manufacturer’s guidelines.

Samples from each of the patients were tested for CD diagnosis by antibodies against IgA-TG2 (human recombinant transglutaminase, EliA Celikey IgA Kit ThermoFisher Scientific, Waltham, MA, USA) and IgG-DGP (human IgG antibodies, EliA IgG conjugate) by means of an enzyme-linked immunosorbent assay (EliA). Both tests were performed with the automated fluoroenzyme INMUNOCAP 250 instrument (Thermo Fisher Scientific) in accordance with the manufacturer’s instructions.

For anti IgA-TG2, the results were considered to be positive and negative at IgA-TG2 >8 U/mL or <2 U/mL, respectively. Between 2 and 8 (U/mL), considered to be a grey zone, a follow-up test was established. These biological intervals were established by clinical consensus in our pediatric Spanish population based on previous studies [1,14].

Antibody levels were calculated in units per milliliter using a 6-parameter standard curve, as provided by the manufacturer (range between 0 and 128 U/mL), and we calculated the exact value of IgA-TG2 if the antibody concentrations were above the measurement range. The sera were serially diluted, and the values were corrected by the dilution factor.

The pipettes used for the dilutions followed ENAC’s metrological traceability policy to ensure the validity of the results, and were verified and calibrated in compliance with ISO 8655 standards, which was carried out by a laboratory accredited by AENOR and ENAC for ISO 17025.

For antibodies against synthetic IgG-DGP, the results were reported as units, where values <7 U/mL (manufacturer’s guidelines) were considered to be normal. IgA-TG2 and IgG-DGP were tested in one step in children under 4 years of age or in the case of children ≥4 years old with IgA deficiency.

IgA-EMA testing was performed by indirect immunofluorescence (IFI) (AESKU slides, AESKU-Diagnostics, Wendelsheim, Germany) that used sections of distal monkey esophagus as a substrate. EMA assays were read by 2 experienced observers. A titer of 1:10 was reported to be the threshold for positivity, as established by the manufacturer, and positive samples were serially diluted from titers of 1:10 to 1:2560. IgA-EMA antibodies were measured in all samples with IgA-TG2 values > 2 U/mL.

Positive and negative control samples were analyzed in each run, and the laboratory had successful participation in interlaboratory comparison programs for analysis and interpretation of the results: UK NEKAS (SEQC) United Kingdom and the Quality Club (ThermoFisher, Friburgo, Germany).

### 2.3. Genotyping

HLA genotyping was performed using the single specific primer polymerase chain reaction (SSPPCR) DQ kits DQA1*05, DQB1*02, DQA1*0301, DQB1*0302, DQA1*0505, DQB1*0202for detecting the DQ2.5, DQ2.2, and DQ8 haplotypes (Celiacstrip HLA DQ2DQ8 OPERON, Inmuno and Molecular Diagnostics, Zaragoza, Spain).

### 2.4. Histopathological Evaluation of Intestinal Mucosa

Intestinal biopsies were obtained from all patients who did not meet the minimum ESPGHAN criteria for avoiding biopsy. At least 4 endoscopy biopsy samples of each patient, including two samples from the bulbus, were taken. Lesions were graded according to the Marsh–Oberhuber classification [15] by an experienced pathologist.

### 2.5. CD Follow-Up

Patients were followed from the time of diagnosis for 24 months. At follow-up, clinical and IgA-TG2 evaluations were made.

### 2.6. Ethical Considerations

The study followed the ethical guidelines given in the Declaration of Helsinki, and the institutional review board approved the study protocol, registered as code number 2019/098. The data analyzed in this study have been handled in accordance with current legislation and the procedures established by the ethics commission to ensure the protection of personal data in accordance with the General Data Protection Regulation (Regulation (EU) 2016/679) and Organic Law 3/2018.

### 2.7. Statistical Analysis

The data are summarized with absolute and relative frequencies. The non-normality of the data led us to use synthesis statistics such as the median and interquartile ranges. A non-parametric test (Kruskal–Wallis) was used to compare EMA and IgA-TG2 median by age groups. Differences were considered statistically significant at *p* < 0.05. The calculations were carried out with SPSS 22.0.

## 3. Results

The flowchart (Figure 1) shows the results of applying the 2012 ESPGAN criteria to the whole cohort.

A diagnosis of CD was confirmed in 113 patients (5.72 ± 4.62 years old; 53% boys and 47% girls) from 5641 patients with suspected CD. Confirmation of the diagnosis was made by serologic tests in most patients.

Small bowel biopsy (SBB) was omitted in 84.1% of the symptomatic patients with CD.

Regarding clinical presentation, most patients (61.95%) presented with classical intestinal manifestations. With respect to extra-intestinal manifestations, iron deficiency microcytic anemia was the most frequent (25.66%). CD was confirmed in 9.74% of patients with risk factors (CD in a first-degree family member, Type I diabetes, autoimmune thyroid disease, and autism) (Table 1).

### 3.1. Serological CD Diagnosis

With respect to the serological results, it was highlighted that:100% of pediatric patients for whom IgA-TG2 ≥ 10 ULN (80.53%) were IgA-EMA-positive.Furthermore, we observed a positive correlation between IgA-TG2 titers ≥ 10 ULN and strong EMA positivity (*p* < 0.001), as shown in Figure 2. Above an average value of ≥10 ULN (80 U/mL) of IgA-TG2, the IgA-EMA titer was greater than 1:80, even in asymptomatic patients with extra-intestinal symptoms or belonging to the at-risk group.We found a negative correlation between age and IgA-TG2 (*p* < 0.001), with higher values in children aged <3 years old (Figure 3).

### 3.2. Genotyping

On the other hand, 105 (92.96%) patients were tested for HLA. Of the patients 82.9% carried HLA-DQ2.5 encoded by the DQA1∗05 (alpha chain) and the DQB1∗02 (beta chain) genes (Figure 1).

### 3.3. Histopathological Evaluation of Intestinal Mucosa

Biopsies were needed in 15.9% of patients (Figure 1). Of these, six had confirmed CD and two did not have CD. Most CD patients (72.1%) had villous atrophy Marsh III. The values of IgA-TG2 in patients who avoided biopsy are shown in Figure 4.

### 3.4. Serologic Response to a GFD

All symptomatic patients had clinical and serological improvements after a GFD, showing decreasing titers of IgA-TG2 over time, with the highest decrease at 6 months of the GFD. After 24 months of follow-up, 95.5% of patients showed negative IgA-TG2 titers (Figure 1), but 4.5% patients who did not improve showed voluntary and involuntary dietary transgression.

## 4. Discussion

Optimizing CD diagnostics requires continuous investigation [2]. Some studies have considered IgA-TG2 as a good alternative to IgA-EMA [7,16]. IgA-TG2 is the first-line recommended serologic test for CD screening in individuals aged >2 years, with high specificity and sensitivity (above 95%) [3,17]. High levels of IgA-TG2 (≥10 ULN (upper linearity limit)) are a reliable and accurate test for diagnosing CD [10]. In agreement with these aspects, we used the IgA-TG2 test as first-line serologic test, obtaining the results previously cited. In our study, at IgA-TG2 ≥ 10 ULN (upper linearity limit), we performed serial dilutions and we obtained quantifications of very high values of IgA-TG2 (our range in the study was 0.1–6300 U/mL). The use of this dilution technique has been useful in the diagnosis and follow-up of patients, as described below:According to the literature [10,11,18,19], we used IgA-EMA as a confirmatory test. Our results showed that IgA-EMA levels were significantly correlated with IgA-TG2 when the levels are ≥10 ULN.IgA-TG2 is the most widely used method for monitoring dietary adherence [20,21]. We verified that it was normalized in all patients on follow-up.This biomarker measured with this precision provides information on CD and gives us greater a guarantee in the follow-up of the response to the GFD, with a minimum cost.

We have no knowledge of studies that have provided IgA-TG2 titers with this accuracy. We think that the correct dilution provides a safety measure in diagnosis and follow-up, and allows us to exactly know the serological response to the GFD.

In IgA-sufficient patients, IgA-TG2 is the most predictive and reproducible single test. In the case of pediatric patients <2 years old, who had IgA levels physiologically low), we used both IgA-TG2 and IgG-GPD; using the recommendations of previous studies [10,12,22].

The ESPGHAN guidelines were proposed to avoid biopsies without affecting diagnostic accuracy [10,11]. In our study, 5641 pediatric patients with suspected CD were enrolled. CD was confirmed in 113 patients, without SBB in most patients (84.1%). These biopsy-sparing data are higher compared with those of previous reports. Worldwide, there are some studies, both retrospective [19,23,24,25,26] and prospective [4,27,28,29,30,31,32]. According to the first one, the rate of biopsies avoided varies from 52% (26) to 77.4% [24]. Regarding later studies, Samarazo et al., concluded that CD diagnosis still relies on SBB when tests are not available [29]. Our data on avoided biopsies are higher because our laboratory has easy access to reliable serologic testing for EMA and genetic tests.

If we consider that biopsy represents invasive and expensive surgical procedure [4] and that the mean age at CD diagnosis is 4 years old [29], biopsies should be reduced as much as possible. In our study, the mean age was 5 years. On the other hand, 41.6% of all pediatric patients are <4 years, so avoiding biopsies could reduce the morbidity of this procedure and additionally reduce costs. Another aspect we should take into consideration is that the younger the patient is, the deeper the sedation we have to use for performing an intestinal biopsy, including general anesthesia [33].

IgA EMA is used as a confirmatory test [17,34]. Recent studies have shown that EMA should not be routinely tested in screening (alone or with IgA-TG2) but should only be used as a confirmatory test in case of an uncertain diagnosis (weak positivity in high-risk populations) [35]. Mursh et al. [16], in the ESPGHAN guidelines, proposed that if EMA antibody testing is not locally available, a second positive IgA-TG2 antibody may be substituted and the serum saved for later EMA testing. In a prospective study, Mubarak concluded that SBB is not necessary for a diagnosis of CD in symptomatic patients with IgA-TG2 ≥ 100 U/mL [36].

In our study, we verified that IgA-EMA levels are significantly correlated with IgA-TG2 when they exceed ≥10 ULN. Furthermore, in our study, quantitative IgA-TG2 was compared with IgA-EMA. When IgA-TG2 was above the detection threshold, as a result, we obtained a significant correlation between both levels. Thus, when IgA-TG2 is ≥10 ULN, IgA-EMA is not necessary. To our knowledge, this is the first study to offer such data (an IgA-TG2 range of 0.1–6200 U/mL and IgA-EMA titers from 1:10 to 1:2560) as a possible simplification of the ESPGHAN criteria. This aspect is of special interest because:Not all countries have the necessary resources and facilities to analyze IgA-EMA. One study performed in 13 Mediterranean countries (western and eastern WHO regions, including Spain) [29] enrolled 1974 pediatric patients and confirmed CD in 25.9% of them. Nevertheless, only 14 patients were diagnosed of CD by the 2012 ESPGHAN guidelines. In that study, 40 Spanish pediatric patients were screened for CD and only three patients could be diagnosed by the ESPGHAN criteria.The IgA-EMA test is relatively expensive and requires experienced observers for microscopic evaluation, with potential inter-observer variation.IgA-EMA is detected on monkey esophagus by immunofluorescence, which raises ethical concerns related to the use of endangered species as a substrate.

We have also linked the values of IgA-TG2 with the titers IgA-EMA values. This has been shown to have a specificity of nearly 100% for the diagnosis of CD. False positives for anti-IgA-TG2 normally display a low antibody titer (less than twice the cut-off) [1]. Gidrewicz et al. [19] highlighted the strength of EMA testing in low IgA-TG2 titers; these authors said that EMA testing improves the PPV when the IgA-TG2 is low or moderately elevated, and concluded that an EMA ≥1:80 achieved 100% PPV [19]. A positive EMA result as a mandatory criterion for the no-biopsy approach is still being debated. Moreover, Husby considered that in IgA-sufficient patients, IgA-TG2 is the most predictive and reproducible single test [10], although IgA EMA performs similarly well in some expert laboratories and is used as a confirmatory test. These observations are related to our results that show a 100% concordance between the EMA titer and the IgA-TG2 values measured on quantitative and qualitative value scales.

This study provides evidence of a positive correlation between IgA-TG2 antibody serum levels and IgA-EMA. We obtained 100% concordance between IgA-TG2 ≥ 10 ULN and positive results for IgA-EMA; furthermore, we have observed a correlation between the quantitative values of IgA-TG2 U/mL and IgA-EMA antibody titers (*p* < 0.001; Figure 2). If EMA testing was not available a diagnosis could be guaranteed by both IgA-TG2 values ≥ 10 ULN (on second test) and serological tests after GFD. It is possible to use IgA-TG2 as a marker in two different samples, and we suggest that IgA- EMA would be required only for patients with lower titers or discordant antibody results.

Husby et al. [34], in the recommendations of the American Gastroenterological Association (AGA), suggested that HLA-DQ2/DQ8 determination has a limited role in the diagnosis of CD. Its value is largely related to its negative predictive value to rule out CD in patients who are seronegative. Werkstetter [4], in a large prospective study with 645 patients, concluded that is not required for an accurate diagnosis. Clouzeau-Girard et al. [37] showed that determination of the HLA-DQ2 and HLA-DQ8 status can be used to support the diagnosis of CD, but it is not an essential condition to confirm a diagnosis. CD is unlikely when HLA is negative. Gidrewicz et al. [19], in a study with 775 patients with positive IgA-TG2, showed that genetic susceptibility testing is not helpful for identifying false positive patients.

The ESPGHAN guidelines of 2012 included testing HLA-DQ2/DQ8 for a no-biopsy diagnosis, but the new ESPGHAN 2020 recommendations suggested that HLA typing is not required in patients with positive IgA-TG2 (≥10 ULN) with a confirmation of being IgA-EMA-positive. We studied 105 cases of 113 patients enrolled in our study, and seven of them had no DQ2/DQ8, five had one of the two alleles, but two (a girl and a boy) did not have any risk alleles; however the values of IgA-TG2 were high, especially in one of them, who had 77 * ULN (620 U/mL) and presented both classical symptoms and a response to the GFD. The family rejected the biopsy due to clinical and serological improvement.

Laboratory tests and biochemical evaluation are crucial to verify the accuracy of the diagnosis. A key issue in the follow-up is the usefulness of serology and whether a decline in antibody levels is sufficient evidence for proper management [37]. Patients should be monitored regularly to check the normalization of antibody determination and clinical normalization [22]. Children should be followed up after 4–6 months from diagnosis and then every year to check symptomatic improvement, adherence to the GFD, quality of life, and progressive normalization of CD-related antibodies [1]. How long it takes for antibody titers to normalize depends on the initial level, although it is generally achieved within 6–12 months after starting the gluten-free diet [21].

IgA-TG2 is the main marker for diagnosis and follow-up of a response to GFD. We consider it important to give the exact quantitative result, especially in children whose debut levels of this marker are often very high; after dilution, we can obtain the exact value needed to verify that the diet has been properly well applied. We have found that the concentration of IgA-TG2 decreases rapidly after establishment of the GFD, with a decrease of more than 60% compared with the initial values. However, 30% of children, who presented with very high concentrations at diagnosis, had values above the upper linearity limit (>128 U/mL). Therefore, we highlight the use of the exact values in diagnosis in order to be certain of the decrease in values, especially in less oligosymptomatic or subclinical cases.

We observed that children showed an adequate adherence to the GFD and showed decreasing celiac antibody levels.

In this study, it was shown that 95/95 patients with IgA-TG2 ≥ 10 ULN suffered from CD (VPP 100%). In all cases, IgA-EMA antibodies were positive and their titers were directly proportional to the value of ULN. All the patients were confirmed to have normalization of the values after the GFD. Therefore, we consider this test to be redundant in these cases, and it would be preferable for confirmation in cases with values less than 10 ≥ ULN.

In this study, with a large sample in relation to the studies by other authors, we have verified that further simplifications of the ESPGHAN guidelines might be made.

## 5. Conclusions

The chance of avoiding a small intestinal biopsy and EMA test is based on the strength of the correlation between the IgA-TG2 and the stage of mucosal damage. These aspects overcome the disadvantages of biopsy in pediatric patients (invasive technique) and of the IgA-EMA test (cost, not sustainable).

## Figures and Tables

**Figure 1 ijerph-19-05020-f001:**
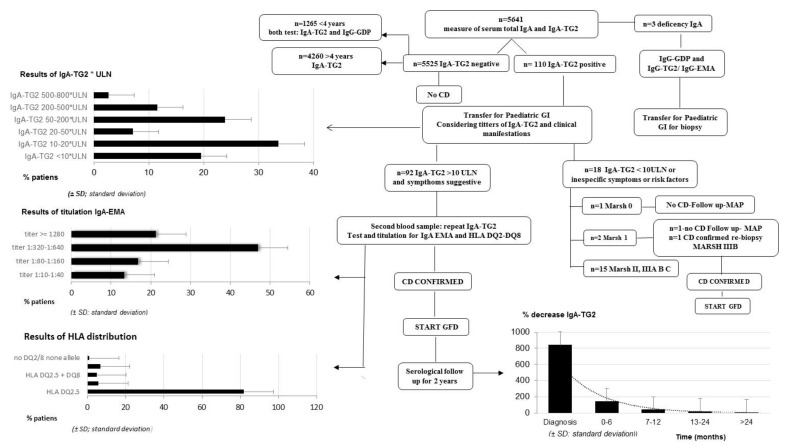
CD diagnosis process with main results. Abbreviations: CD, celiac disease; ESPGHAN, European Society for Pediatric Gastroenterology, Hepatology, and Nutrition; GFD, gluten-free diet; GI, gastroenterologist; HLA, human leukocyte antigen; IgA, immunoglobulin A; IgA-EMA, IgA anti-endomysial antibodies; IgA-TG2, IgA antibodies against tissue transglutaminase Type 2; IgG, immunoglobulin G; IgG-GDP, IgG antibodies against deamidated gliadin peptides; PAD, primary attention doctor; ULN, upper limit of normal.

**Figure 2 ijerph-19-05020-f002:**
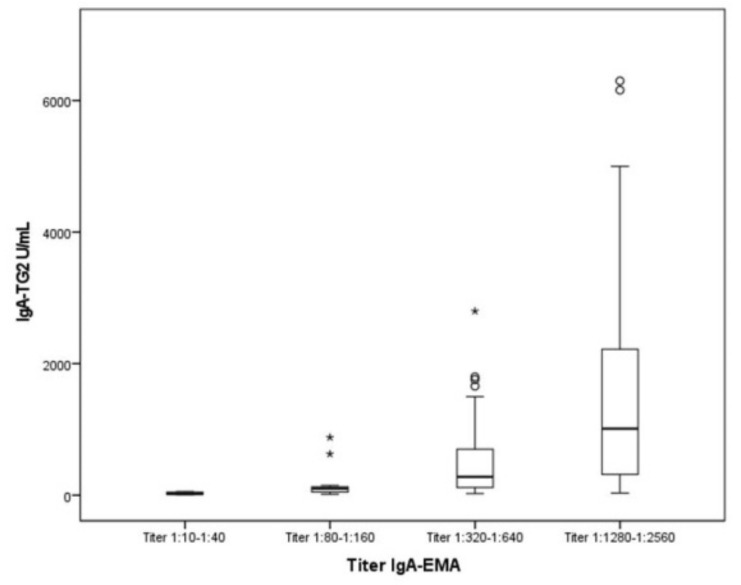
Relationship between IgA-TG2 and EMA titers. A non-parametric test comparing the medians among four groups of EMA. *p* < 0.001. Abbreviations: IgA-EMA, IgA anti-endomysial antibodies; IgA-TG2, IgA antibodies against tissue transglutaminase Type 2. ° Outliers are values between the IORs and three IORs from the end of a box. * Values more than three IORs from the end of a box.

**Figure 3 ijerph-19-05020-f003:**
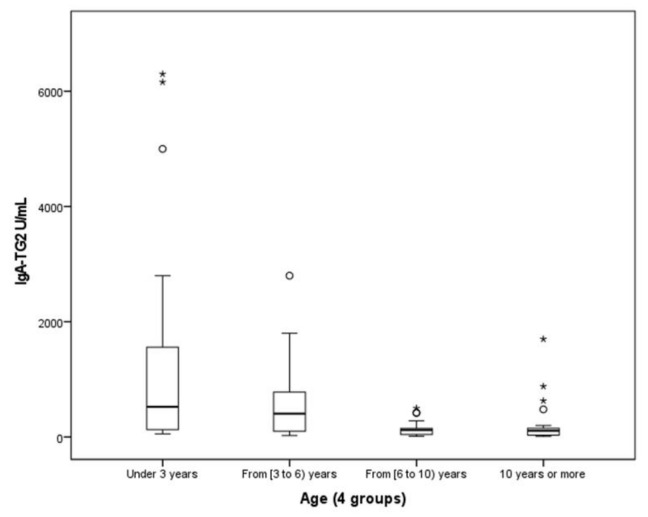
Relationship between age and IgA-TG2. Abbreviations: IgA-TG2, IgA antibodies against tissue transglutaminase Type 2. °, Outliers are values between IORs and three IORs from the end of a box. * Values more than three IORs from the end of a box.

**Figure 4 ijerph-19-05020-f004:**
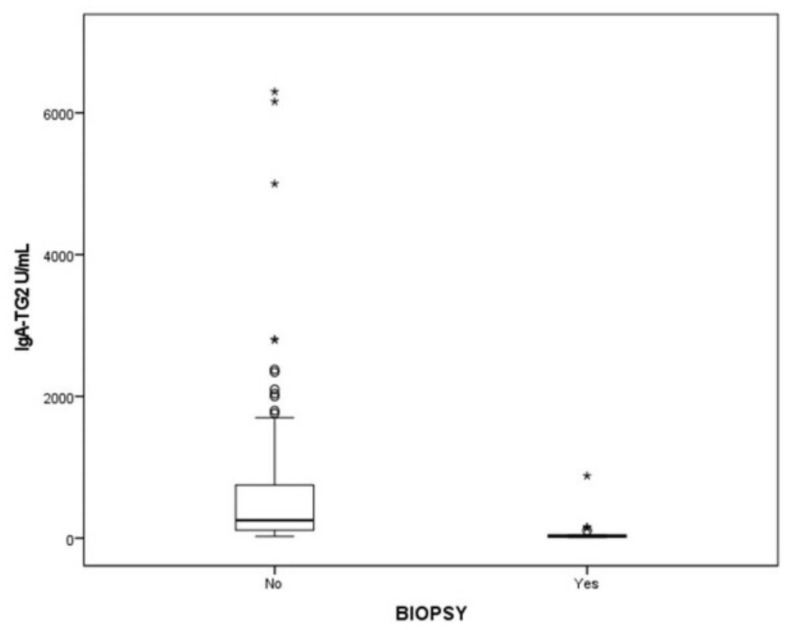
Values of IgA-TG2 (U/mL) in patients who avoided biopsy or underwent it. Abbreviations: IgA-TG2, antibodies against tissue transglutaminase Type 2. °, Outliers are values between IORs and three IORs from the end of a box. * Values more than three IORs from the end of a box.

**Table 1 ijerph-19-05020-t001:** Clinical onset and risk factors of CD.

		N^o^ Cases	Percentage of Cases
		70	61.95
	Chronic or intermitent diarrhea *, weight loss, and/or growth failure * (**Classic Triad**)		
	Gastrointestinal symptoms *	35	30.97
**Clinical symptoms**	Recurrent abdominal pain, vomiting, chronic, or recurrent constipation		
	Extraintestinal symptoms		
	Chronic iron deficency, anaemia *	29	25.66
	Subclinical hypothyroidism	2	1.77
	Arthritis/arthralgia	1	0.88
	Irritability, chronic fatigue	5	4.42
	Neuropaty, TDHA	2	1.77
	Arthritis/arthralgia	2	1.77
	Recurrent aphthous	1	0.88
	Oligo symtomatic or subclinical	6	5.31
	CD first-degree family member	5	4.42
	Type I diabetes mellitus	1	0.88
**Risk factors**	Autism	1	0.88
	IgA deficiency	3	2.65
	Autoinmunity thyroid disease	1	0.88

* Common symptoms.

## Data Availability

Restrictions apply to the availability of these data (3rd party data).

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
