# Peer review of "Evaluation of Celiac Disease by Minimally Invasive Biomarkers in a Spanish Pediatric Population"

_ijerph, 2022, doi:10.3390/ijerph19095020_

Round 1
Reviewer 1 Report
In this manuscript, the authors have tried to evaluate some biomarkers for diagnosing Celiac disease (CD). This study would provide an important contribution to medical scene relating CD. The topic of this paper is valuable, most importantly, this study represents a massive effort and it would deserve to be published somewhere. In my opinion, the authors should improve the following points.
- The word "non-invasive" in the title of the paper is incongruous. This study examines markers in serum, but I don't think blood sampling is non-invasive.
- In Abstract, line 33, what is “the DSG”? The same word can be found in line 300.
- In Fig.1, the authors should add error bars (± SD; standard deviation) to each graph. In addition, Fig.1 overall, the lines and arrows should be carefully placed (there are some displacements in position) and the colors should be consistent.
- Line 202 and 204, the authors describe about Table 2, but where is Table 2 itself?
Since this study have been conducted in accordance with 2012 ESPGHAN Guidelines, I feel that any extra experimentation beyond the guidelines would be unnatural in this paper. Therefore, I have limited my comments to the readability and clarity of this paper. Thank you for giving me the opportunity to review this paper.
Author Response
Response to Reviewer 1 Comments
Thank you very much for your considerations.
We assume your considerations and incorporate the corresponding changes in the text. The paper with the changes is attached.
Point 1: The word "non-invasive" in the title of the paper is incongruous. This study examines markers in serum, but I don't think blood sampling is non-invasive..
Response 1: We change “non-invasive” by “minimally invasive”.
Point 2. In Abstract, line 33, what is “the DSG”? The same word can be found in line 300. Response 2. This is a mistake.“DSG is changed by “GFD”. This abreviation means gluten free diet.
Point 3. In Fig.1, the authors should add error bars (± SD; standard deviation) to each graph. In addition, Fig.1 overall, the lines and arrows should be carefully placed (there are some displacements in position) and the colors should be consistent.
Response 3. We add error bars (± SD; standard deviation) to each graph and change the lines and arrows as well as colors.
Point 4. Line 202 and 204, the authors describe about Table 2, but where is Table 2 itself?
Response 4. This is a mistake.“ Table 2” is eliminated.

Reviewer 2 Report
Authors share their experience in the use of laboratory tests in the diagnosis of celiac disease in children, applying the ESPGHAN 2012 diagnostic criteria. They also propose some recommendations in the use of serologic tests, and how might be restricted the use of IgA-EmA. Their results seems to support the elimination of the determination of risk HLA-DQ alleles proposed in ESPGHAN 2020 criteria, to avoid intestinal biopsy. Perhaps, they should show this fact more clearly and discuss it more extensively.
As a whole, the work is interesting an easy to read, although some opinions are debatable.
Some issues to improve or edit:
-Abstract, lines 26-27: "CD diagnose was confirmed in 113 26 patients, 111 were IgA-TTG positive. 3 (2.7%) had IgA deficiency.": The sum of patients is 114. In flowchart (fig 1) appear 110 IgA-TTG positive and 3 IgA deficiencies.
-Antitransglutaminase ab. are written as TTG and TG2: only one acronym should be used.
-Methods: in the description of HLA-DQ genotyping test, the reference to HLA-DQA1 alleles is missed.
-It is shown one case of CD patient carrying no HLA-DQ risk allele. In an strict use of ESPGHAN 2012 criteria, this patient should have been biopsied. This case should be discussed.
-The correlation of titers between IgA-aTG2 and IgA-EmA is well known. The aim of validating IgA-aTG2 with IgA-EmA is detecting false positive cases of IgA-aTG2. These cases are spare, but they exist (and not always with low values). Perhaps the casuistic presented is too short to find one of these cases. To gain efficiency, it might be more useful do not to titrate IgA-EmA, as the quantification is provided by IgA-aTG2.
-From my view, to test dilutions of sera with high values of IgA-aTG2 is an unnecessary expense when the upper limit of linearity is far higher than 10xULN, and the aim of GFD is to reach negative values. However, it is a good matter to discuss.
-The acronym for intestinal biopsy is used as SSB and SBB.
-Discussion, line 337: "detection limit (> 128 U / mL).": it should be "upper linearity limit".
Author Response
Response to Reviewer 2 Comments
Thank you very much for your considerations.
We assume your considerations and incorporate the corresponding changes in the text. The paper with the changes is attached.
Point 1: Abstract, lines 26-27: "CD diagnose was confirmed in 113 26 patients, 111 were IgA-TTG positive. 3 (2.7%) had IgA deficiency.": The sum of patients is 114. In flowchart (fig 1) appear 110 IgA-TTG positive and 3 IgA deficiencies.
Response 1: This is a mistake. We change “111” by “110”.
Point 2. Antitransglutaminase ab. are written as TTG and TG2: only one acronym should be used.
Response 2. “TTG” is changed by “TG2”.
Point 3. Methods: in the description of HLA-DQ genotyping test, the reference to HLA-DQA1 alleles is missed.
Response 3. The CeliacStrip test allows the determination of the main HLA haplotypes associated with celiac disease. The test detects the presence or absence of the haplotypes that code for HLA-DQ2 and HLA-DQ8.
|
HLA - DQ2 cis |
DQA1*05 - DQB1*02 - DRB1*03 |
|
HLA - DQ2 trans |
DQA1*05 - DQB1*0301 - DRB1*11/DRB1*12 - DQA1*02 - DQB1*02 - DRB1*07 |
|
HLA - DQ8 |
DQA1*03 - DQB1*0302 - DRB1*04 |
Point 4. It is shown one case of CD patient carrying no HLA-DQ risk allele. In an strict use of ESPGHAN 2012 criteria, this patient should have been biopsied. This case should be discussed.
Response 4. The patient (2014) presented severe symptoms (celiac habit and anemia) IgA-TG2 620 U/mL and IgA-EMA titer 1:1280. Due to the seriousness, the GFD was established before the HLA results were available. The clinical and serological improvement was spectacular and although the biopsy was proposed, the family rejected it.
Point 5. The correlation of titers between IgA-aTG2 and IgA-EmA is well known. The aim of validating IgA-aTG2 with IgA-EmA is detecting false positive cases of IgA-aTG2. These cases are spare, but they exist (and not always with low values). Perhaps the casuistic presented is too short to find one of these cases. To gain efficiency, it might be more useful do not to titrate IgA-EmA, as the quantification is provided by IgA-aTG2.
Response 5. 5641 patients have been studied, there was no case >10xULN with negative IgA-EMA. False positives were more likely for IgA-TG2 values between 1 and 10xULN. Thus, IgA-EMA was very useful to confirm positives with IgA-TG2 values <10xULN, prior to performing the biopsy, or seronegatives with high clinical suspicion. In any case, in a false positive >10xULN the marker would not decrease after GFD.
EMA titration is performed to demonstrate that there is a significant correlation at high values between a quantitative and qualitative test that measures the same antigen, which supports the theory that at IgA-TG2 values >10ULN the IgA-EMA is a redundant test.
Point 6. From my view, to test dilutions of sera with high values of IgA-aTG2 is an unnecessary expense when the upper limit of linearity is far higher than 10xULN, and the aim of GFD is to reach negative values. However, it is a good matter to discuss.
Response 6. Patients with very high IgA-TG2 values (range 1.2-6300 U/mL), may continue with values above the linearity limit in the first review (30% in our series). The exact value of TG2 provides us the value of the decrease in this marker and, therefore, the certainty of the response to the GFD and the correct diagnosis.
In the ESPGHAN 2012 guidelines, monitoring the disease was based on two aspects: clinical improvement and decrease in IgA-TG2 values. With the new ESPGHAN 2020 guidelines, in which biopsy can be omitted in asymptomatic patients, only the decrease in TG2 shows the response to treatment. Therefore, knowing the exact value of this marker is essential.
In our laboratory we have protocolized dilutions: when the IgA-TG2 value is >128 U/mL, we carry out a first 1:20 dilution (68 patients in this study) with which linearity is achieved up to 2560 U/mL. In the case of higher values, a 1:50 dilution is processed with linearity up to 6400 (6 patients in this study). With this procedure, we have achieved the exct value for the follow-up of all patients with minimal cost.
Point 7. The acronym for intestinal biopsy is used as SSB and SBB.
Response 7.“SSB” is changed by “SBB”.
Point 8. Discussion, line 337: "detection limit (> 128 U / mL).": it should be "upper linearity limit".
Response 8."detection limit” is changed by "upper linearity limit".
